**www.cambridge.org/ext**

behaviour; biophilia; biophobia; extinction of experience; shifting baselines

**Corresponding author:**
Kevin J. Gaston;
Email: k.j.gaston@exeter.ac.uk

# Personalised ecology and the future of biodiversity

Kevin J. Gaston[1] [ID], Benjamin B. Phillips[1] and Masashi Soga[2]

[1]Environment and Sustainability Institute, University of Exeter, Penryn, UK and [2]Graduate School of Agricultural and Life Sciences, The University of Tokyo, Tokyo, Japan

## Abstract

The future of biodiversity lies not just in the strategies and mechanisms by which ecosystems and species are practically best protected from anthropogenic pressures. It lies also, and perhaps foremost, in the many billions of decisions that people make that, intentionally or otherwise, shape their impact on nature and the conservation policies and interventions that are implemented. Personalised ecology – the set of direct sensory interactions that each of us has with nature – is one important consideration in understanding the decisions that people make. Indeed, it has long been argued that people's personalised ecologies have powerful implications, as captured in such concepts as biophilia, extinction of experience and shifting baselines. In this paper, we briefly review the connections between personalised ecology and the future of biodiversity, and the ways in which personalised ecologies might usefully be enhanced to improve that future.

## Impact statement

Protecting and restoring nature depend on understanding the billions of decisions that people make. Such decisions range from simple acts like caring for wildlife in one's garden to more complex decisions like what products to buy or which political candidate to support. These decisions are determined in part by direct experiences of, and relationships with, nature. These may affect nature directly (e.g., determining how much an individual disrupts wildlife habitats) or indirectly (e.g., affecting one's thoughts and attitudes toward nature). Understanding how people's relationships with nature differ, how they are changing and how they relate to people's pro-nature attitudes and behaviours can help to reveal strategies that can benefit biodiversity. For example, people who feel more connected to nature are more likely to take action to help protect it. People's relationships with nature might be improved, for example, by increasing the availability and accessibility of natural environments, and people's inclination, ability and confidence to engage with nature. Such efforts have the potential to create a virtuous cycle of human–nature interactions, whereby increased engagement with nature leads to greater appreciation, enjoyment and desire to protect it. This is particularly important at a time when people's relationships with nature are often declining.

## Introduction

Scientific discussion of how to slow and reverse global biodiversity loss has concentrated far more on ecological solutions than on social change. This has been exemplified by papers published in the run-up to the Fifteenth Conference of the Parties (COP-15) to the UN Convention on Biological Diversity (CBD), where research focused strongly on how best to set targets and measure progress for conservation (e.g., Watson et al., 2020; Obura et al., 2021; Allan et al., 2022; Leadley et al., 2022), the importance and maintenance of wilderness areas (e.g., Aycrigg et al., 2022; Pérez-Hämmerle et al., 2022), identifying priority areas for biodiversity conservation, and for expanding, and increasing the effectiveness of, protected area systems (e.g., Hanson et al., 2020; Ward et al., 2020; Allan et al., 2022; Brennan et al., 2022; Wauchope et al., 2022) and understanding the threats to, and recovery of, individual species (e.g., Grace et al., 2021; Mair et al., 2021; Bolam et al., 2022). There is no doubt that these are all vitally important issues. However, the loss of biodiversity has been an outcome of many billions of decisions (with varying degrees of independence) by individual people. Such decisions, intentional and otherwise, include how people use and manage any natural resources that they have direct influence over (from domestic gardens and backyards to larger land and sea holdings), what resources and items they purchase as consumers, what organisations they encourage and assist (e.g., conservation NGOs), and which local, regional and national governmental policies and management interventions they support. These pathways, and how they can best be influenced, have long been

studied within environmental sustainability. They have, however, attracted far less attention from the biodiversity conservation community.

This is not to say that behavioural decision making and social change have received no attention in the context of biodiversity conservation (Thomas-Walters et al., 2023). Interest has included such issues as managing demand for wildlife products (MacFarlane et al., 2022), the promotion of farmers' pro-environmental practices (Mastrangelo et al., 2014), the application of 'nudge theory' (Nelson et al., 2019), conservation messaging (Kidd et al., 2019), the effectiveness of social marketing campaigns (Green et al., 2019) and the influence of visual media on human–nature interactions (Silk et al., 2021). Nonetheless, it does seem remarkable that, despite being raised at least a decade ago (e.g., St John et al., 2010), it continues to be necessary for recent papers (including in high profile journals) to champion and highlight the role that the behavioural sciences, for example, could play in biodiversity conservation (e.g., Maynard et al., 2020; Balmford et al., 2021; Nielsen et al., 2021).

A range of different viewpoints can help to understand how individual decisions are determined, the negative impacts on biodiversity and ways of reducing these (Clayton et al., 2013; Amel et al., 2017; Reddy et al., 2017; Ives et al., 2018). One is that of personalised ecology, which describes the set of direct interactions that each of us has with nature (Gaston et al., 2018; Gaston, 2020; Soga and Gaston, 2022). Whilst the significance of such interactions, which are likely unique to each person in their composition, has long been recognised (e.g., Wilson, 1984; Kellert and Wilson, 1993; Pyle, 1993; Stokes, 2006; Samways, 2007), it has particularly come to the fore of recent (Clayton et al., 2017; Soga and Gaston, 2022). This paper describes why personalised ecology provides a pertinent perspective by exploring the links with, and implications for, the future of biodiversity. Some of the issues discussed (e.g., biophilia, connection to nature, extinction of experience, shifting baselines) have been argued to be amongst the most vital for that future (e.g., Ehrlich and Kennedy, 2005; Kareiva, 2008; Simaika and Samways, 2010). Given strong biases in the relevant research literature toward studies of culturally westernised societies, our considerations are similarly biased, although many may generalise more widely.

## Personalised ecology

In the most fundamental sense, an individual's personalised ecology describes all of their direct interactions with nature. This includes those with both micro- and macro-organisms. However, a narrower sense conception of personalised ecology, which is of more relevance in the present context, is the direct sensory interactions a person has with nature, predominantly through sight, sound, smell and touch. This is largely with macro-organisms. It is the focus on direct interactions which differentiates personalised ecology from broader considerations of ecosystem services (from which individual people frequently benefit without their necessarily interacting directly with the organisms providing such benefits; Gaston et al., 2018).

Personalised ecology is concerned with interactions with nature. There has long been debate as to where the limits to what constitutes nature should lie and definitions can differ markedly, particularly across cultures and disciplines (e.g., Wohlwill, 1983; Proctor, 1998; Wickson, 2008; Bratman et al., 2012; Hartig et al., 2014; CBD, 2022). We use the same definition here as we have employed in other recent studies about human–nature interactions

(e.g., Soga and Gaston, 2020, 2022), in which nature encompasses individual living organisms through to ecosystems, excluding those that are not self-sustaining. This enables a focus on essentially 'wild' organisms.

People's direct interactions with some species and taxonomic groups have received much attention (e.g., sharks, snakes, bears; Chippaux, 2017; Bombieri et al., 2018; Gibbs, 2021), often due to the perceived, potential or realised negative threat that they pose to people. On the whole, however, personalised ecologies remain poorly documented, and have been studied in relatively crude terms, for example, measuring the extent of greenspace in the vicinity of a person's home or workplace, or the kind, frequency and duration of outdoor visits that they make (e.g., Shanahan et al., 2016; Cox et al., 2017b; White et al., 2019; Colley et al., 2022). Studies of human interactions with other species have almost invariably focused on the numbers and types of species that occur where an individual person lives or visits, rather than considering which ones, and in what numbers, an individual person actually encounters and experiences them (e.g., Fuller et al., 2007; Dallimer et al., 2012; Methorst et al., 2021).

Notwithstanding, it is apparent that personalised ecologies vary greatly amongst individual people, both within and between populations. On average, personalised ecologies will relate to the spatial variation of those components of biodiversity of which people tend to be more aware (e.g., larger organisms). In urbanised societies, and probably more widely, personalised ecologies can be very poor for many people. They are often also highly skewed such that the majority of nature interactions that do occur are experienced by only a small proportion of people (Cox et al., 2017a). In general, personalised ecologies are dependent on opportunity (e.g., the local presence and abundance of species), motivation (e.g., emotional affinity with nature) and capability (e.g., ability to see or hear particular species) (Dallimer et al., 2014; Soga and Gaston, 2022). These are in turn often related to socioeconomic circumstances. We return to these issues in more detail later.

Not only do personalised ecologies vary greatly between people, but an individual's personalised ecology also varies across multiple time scales (Soga and Gaston, 2022). It changes through the day (often peaking when people are moving outdoors; Derks et al., 2020), through the week (often peaking at weekends when people engage more in outdoor recreation; Veitch et al., 2015), and through a person's life course (often peaking both during childhood and during earlier periods of retirement; Hughes et al., 2019). In much of the western world, personalised ecologies, especially those experienced by children, have also become progressively more limited across recent generations, a phenomenon referred to as extinction of experience (Pyle, 1993; Miller, 2005; Soga and Gaston, 2016, 2023). On the other hand, international, and particularly intercontinental, travel has broadened (though not necessarily deepened) the personalised ecologies of a (typically small) minority of many human populations, allowing people to interact with species and ecosystems that are very different from those they would otherwise encounter. This is reflected most strongly through ecotourism.

## Consequences for the future of biodiversity

People's personalised ecologies have a wide array of consequences for the future of biodiversity. Most attention to positive direct interactions with nature has focused on the wellbeing benefits for people, with evidence of impacts on physical, psychological and social health (e.g., Keniger et al., 2013; Hartig et al.,

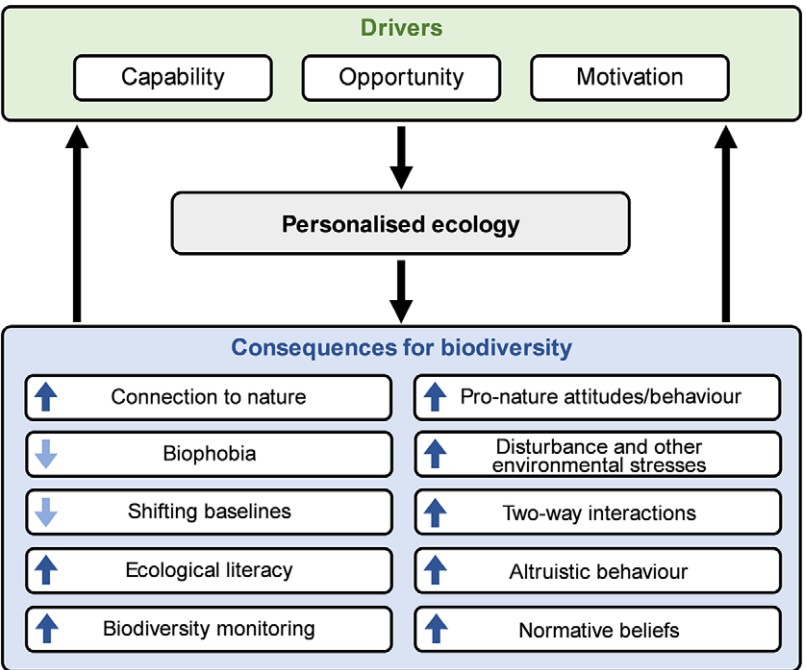

**Figure 1.** A conceptual diagram for understanding the drivers of personalised ecology and its consequences for biodiversity. There is likely a feedback loop in which the consequences of personalised ecology affect its drivers. In the consequences domain of Figure 1, each box contains an up or down arrow that denotes the direction of change in each factor or process caused by increased personalised ecology. For instance, the up arrow in the connection to nature box indicates that direct interactions with nature enhance one's connection to nature.

2014; Bratman et al., 2019; Marselle et al., 2021; Oh et al., 2022). This has led to the development of a diversity of interventions to increase these benefits, focusing largely either on changing the environments in which people spend their time, or on changing their behaviour (Shanahan et al., 2019). The promotion of people–nature interactions for the purpose of improving human wellbeing does, of course, have the potential to benefit biodiversity directly (especially wild plants and animals living in urban areas), including through the creation and maintenance of accessible greenspaces that enable such interactions. However, there are a variety of other consequences of personalised ecologies, both positive and negative, that may have much greater importance for the future of biodiversity (Figure 1).

### (i) Connection to nature

People have a subjective, and perhaps innate, sense of connection with the natural world, sometimes known as biophilia (Wilson, 1984). Such nature connectedness varies dramatically amongst people and societies (Richardson et al., 2022). It is increasingly seen as a core issue in human–nature relationships (Richardson et al., 2020a), and meta-analyses have found that individuals with greater connection to nature have more pro-nature behaviours (Whitburn et al., 2019; Barragan-Jason et al., 2022). The strength of this connection to nature is thought to be enhanced in individuals with a richer and deeper personalised ecology and, conversely, to be weakened in those whose personalised ecology is poorer (Richardson et al., 2020b; Mikołajczak et al., 2021; Li et al., 2022; Lim et al., 2022). Indeed, whilst Wilson (1984) defined biophilia as 'the innate tendency to focus on life and lifelike processes', in subsequent writings he emphasised that it is a complex set of learned behaviours, that is, a disposition that is reinforced,

amplified and expressed through human culture (Wilson, 1993). This is now supported by empirical evidence (Figure 2A; Collado et al., 2013; Vanderstock et al., 2022; Wu et al., 2023). Connection to nature has also been found to mediate the link between personalised ecologies and pro-nature behaviours (Liu et al., 2022).

### (ii) Pro-nature attitudes and behaviours

A key question relating to personalised ecologies is whether they influence a person's pro-nature attitudes and behaviours, which has recently been termed the nature benefit hypothesis (Soga and Gaston, 2022). Several studies have documented positive relationships between levels of nature experience and pro-environmental attitudes and behaviours (Figure 2B; e.g., Wells and Lekies, 2006; Zelenski et al., 2015; Broom, 2017; Rosa et al., 2018; Dean et al., 2019; Alcock et al., 2020; Martin et al., 2020; Liu and Chen, 2021; Ngo et al., 2022). A smaller number have tested for and documented positive relationships for more explicitly pro-nature (a subset of pro-environmental) attitudes and behaviours (Cooper et al., 2015; Soga et al., 2016; Prévot et al., 2018).

### (iii) Biophobia

Whilst reduced positive interactions with nature may weaken support for biodiversity conservation, there is an additional concern that these reductions may strengthen antagonism toward such interactions, that is, a negative feedback loop whereby the less that people interact with nature the less they want to do so. This could occur if the loss of positive nature interactions resulted in an increase in wariness or phobia toward nature, that is, biophobia (Ulrich, 1993). Indeed, there is evidence that extinction of experience is associated with an increase in biophobia, including due to its

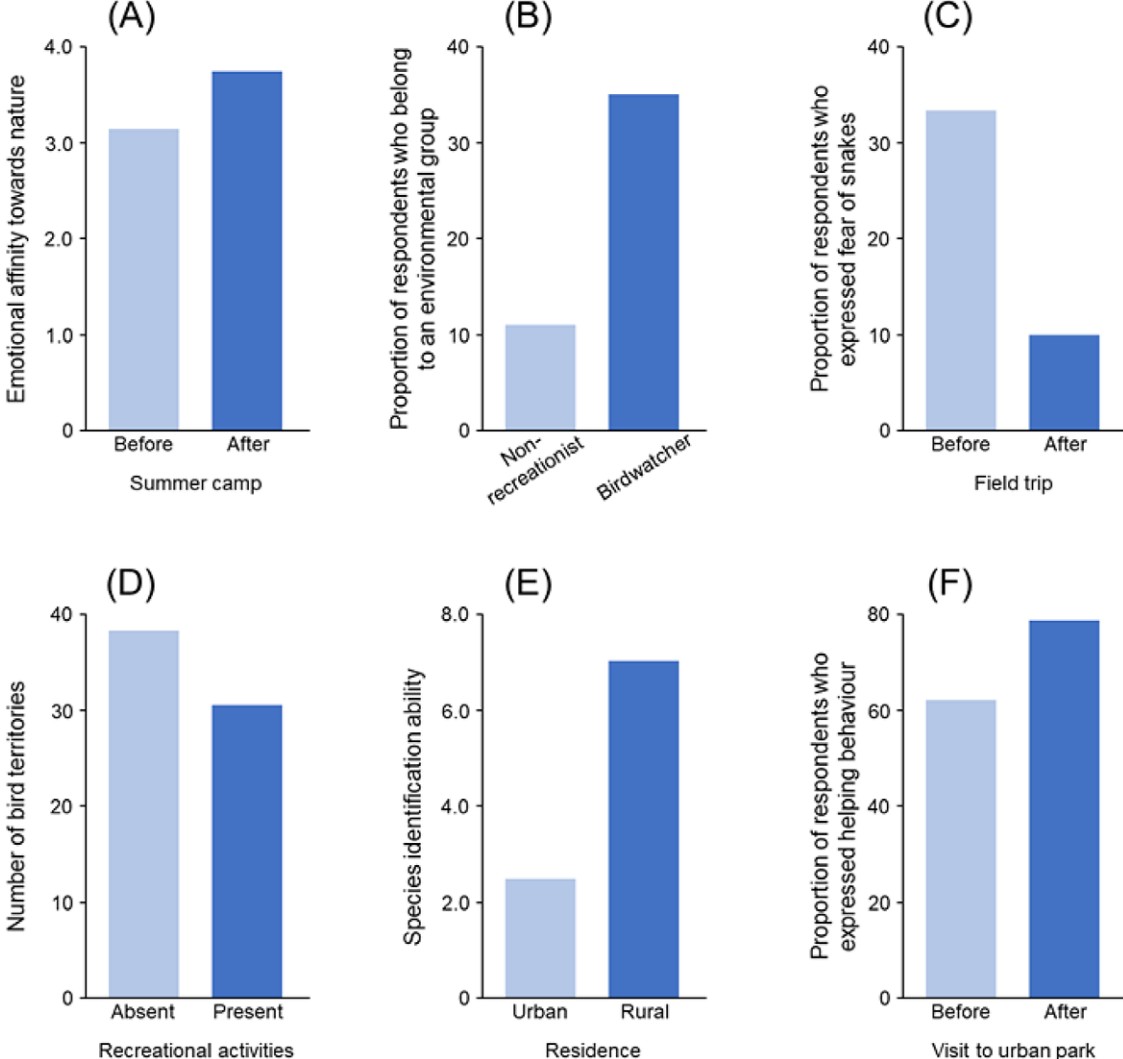

**Figure 2.** Empirical evidence suggesting several possible impacts of increased personalised ecology on biodiversity (A: connection to nature; B: pro-nature attitudes/behaviour; C: biophobia; D: disturbance of wildlife; E: ecological literacy; F: altruistic behaviour). Plots show: (A) changes in emotional connection to nature (measured by the Emotional Affinity toward Nature scale) before and after participating in a nature-based recreational program (summer camps) (Collado et al., 2013); (B) differences in the likelihood of engaging in a pro-nature behaviour between birdwatchers and those who do not use nature for recreational purposes (non-recreationist) (Cooper et al., 2015); (C) changes in the proportion of people exhibiting fear of snakes before and after participating in a field trip (Ballouard et al., 2012); (D) differences in the number of bird territories between sites with and without recreational activities (Bötsch et al., 2017); (E) differences in species identification ability between people who live in urban versus rural areas (Bashan et al., 2021); and (F) changes in the proportion of people exhibiting helping behaviour before and after experience of an urban green park (Guéguen and Stefan, 2014).

associated loss of knowledge about nature (e.g., ability to identify species; Figure 2C; Ballouard et al., 2012; Silva and Minor, 2017; Ngo et al., 2019; Soga et al., 2020; Fukano and Soga, 2021; Sugiyama et al., 2021).

Elevated biophobia can have a wide range of detrimental impacts on the future of biodiversity. Fear emotions impose a significant psychological cost for humans, and thus, increased biophobia can reduce the willingness of local people to coexist with wild animals, particularly, those regarded as dangerous or harmful (e.g., wolves, bears, large cats). Biophobia can therefore decrease public acceptance of certain policies and actions to conserve and restore these organisms (e.g., reintroduction). Biophobia also often results in an increase in persecution of wild organisms, which can negatively impact biodiversity more directly (Pandey et al., 2016; Rocha et al., 2021). If increasing

urbanisation of the human population, and general decline of biodiversity, result in increased biophobia, the impacts on the future of biodiversity could be severe.

### (iv) Negative impacts of nature engagement on biodiversity

Increased human–nature interactions may lead to negative impacts on biodiversity in several ways. This includes, for example, disturbance of wildlife during recreational activities (Figure 2D; Larson et al., 2016; Bötsch et al., 2017), loss of predator avoidance behaviour (Geffroy et al., 2015), the unintentional transport of organisms between sites (including both native and non-native species) (Hodkinson and Thompson, 1997), changes to understorey vegetation (Erfanian et al., 2021), increased chemical pollutants (e.g., negative impacts of sunscreen use on coral reefs; Danovaro et al.,

2008), increased litter and more frequent fires. Better connection to nature might therefore have negative consequences if it means that more people are going out and disturbing or damaging flora and fauna in sensitive areas.

### (v) Two-way interactions

There is evidence of an asymmetry in people's beliefs, whereby they commonly hold that human impacts on the natural environment are greater than the impacts of the natural environment on people (e.g., Coley et al., 2021). This can weaken the role of self, family or human benefits in support for pro-nature behaviours, and is clearly at odds with the utter dependence of humanity on ecosystem services (IPBES, 2019).

### (vi) Shifting baselines

The personalised ecologies that people experience, particularly earlier in life, can have a profound impact on what one regards as 'normal' and 'sound'. Faced with declines in the state of nature, this can result in a progressive 'ratcheting down' or shifting of baselines (Pauly, 1995; Soga and Gaston, 2018). People may, therefore, become more accepting of a much-depleted biodiversity, because the extent of the departure from a natural situation is poorly understood (Jones et al., 2020). Shifting baselines can have many negative impacts on biodiversity conservation as they may lead to an increased tolerance for the progressive degradation of ecosystems, changes in people's expectations as to the state of nature that is worth protecting or restoring, and subsequently the establishment of less ambitious targets and goals for nature conservation (Soga and Gaston, 2018).

### (vii) Ecological literacy

Reduced positive interactions with nature can weaken people's knowledge about local ecosystems (Figure 2E; Bashan et al., 2021). This is often called ecological literacy, or eco-literacy (Pilgrim et al., 2007). It includes, for example, identification skills of local fauna and flora (Bashan et al., 2021), ethnobotanical knowledge (e.g., traditional use of edible/medicinal plants; Okui et al., 2021), and traditional management practices of local ecosystems (Tsuchiya et al., 2014). Maintenance of such knowledge is fundamental for the continued support of local conservation efforts and the capacity of local communities to self-manage natural resources sustainably. Declines in local ecological knowledge can therefore have negative impacts on the conservation of biodiversity.

### (viii) Altruistic behaviour

There is evidence that exposure to nature (e.g., viewing greenery) can contribute to enhancing altruism in humans, which has recently been termed the nature and sustainability hypothesis (Soga and Gaston, 2022). This includes various behaviours, including the reduction of impulsive and selfish decision making and the promotion of sustainable, cooperative and helping behaviour (Figure 2F; Van der Wal et al., 2013; Zelenski et al., 2015; Guéguen and Stefan, 2016). Increased altruistic decision making and behaviour can have a wide range of positive outcomes for biodiversity as those actions can contribute, either directly or indirectly, to the conservation and restoration of wild plants and animals.

### (ix) Biodiversity data collection

Increased nature interactions may, in some cases, contribute to an increased amount of biodiversity data coming from citizens (Schuttler et al., 2018). For example, it has been suggested that increased use of urban greenspaces during the COVID-19 pandemic resulted in increased numbers of wildlife observations submitted to citizen science projects (e.g., Hochachka et al., 2021). Citizen science data can offer a valuable source of species occurrence records and be used to generate species-level information for broad-scale biodiversity mapping and monitoring.

### (x) Normative beliefs

Normative beliefs are the perceptions of what are socially typical or acceptable attitudes and behaviours. For a particular person, both their personalised ecology, and the actual and perceived personalised ecologies of those around them, will shape their normative beliefs. Normative beliefs are often a strong predictor of people's attitudes and behaviours (Armitage and Conner, 2010), including those relating to biodiversity (van Riper et al., 2019). Social norms and normative beliefs may influence attitudes and connection to nature (Oh et al., 2021), or influence behaviour directly in the form of cultural taboos toward the exploitation of particular species, areas and natural resources (Jones et al., 2008), or more subtly through a person's propensity to engage in pro-nature behaviours, such as participation in urban greenspace management (Marshall et al., 2020).

### Acting in combination

Of course, these 10 consequences of personalised ecologies do not act independently, but likely generate a synergistic effect on biodiversity conservation. For example, if people obtain eco-literacy through enhanced personalised ecologies, they may use natural environments in a way that does less harm to those environments (e.g., maintaining appropriate distances from wildlife). Likewise, those with greater species identification ability can provide more accurate and reliable data on biodiversity. Further, increased connection to nature is known to act as a protective factor against biophobia (Zsido et al., 2022). However, there may equally be negative synergies between some of these consequences. For example, shifting baselines are likely to interact with normative beliefs because social norms (perceived or actual) provide a self-sustaining mechanism for maintaining poor personalised ecologies. This may result in a negative, self-reinforcing feedback loop, making it difficult to reverse historic declines in people's personalised ecologies.

### Strengthening personalised ecologies

If better developed personalised ecologies generate positive outcomes for biodiversity conservation, as described above, then strengthening those ecologies may be critically important for the future of biodiversity. People's personalised ecologies can usefully be regarded as being shaped by three broad sets of factors: capability, opportunity and motivation (as distinguished by the COM-B model; Michie et al., 2011). Each of these provides a unique set of opportunities and approaches that could be targeted to strengthen personalised ecologies.

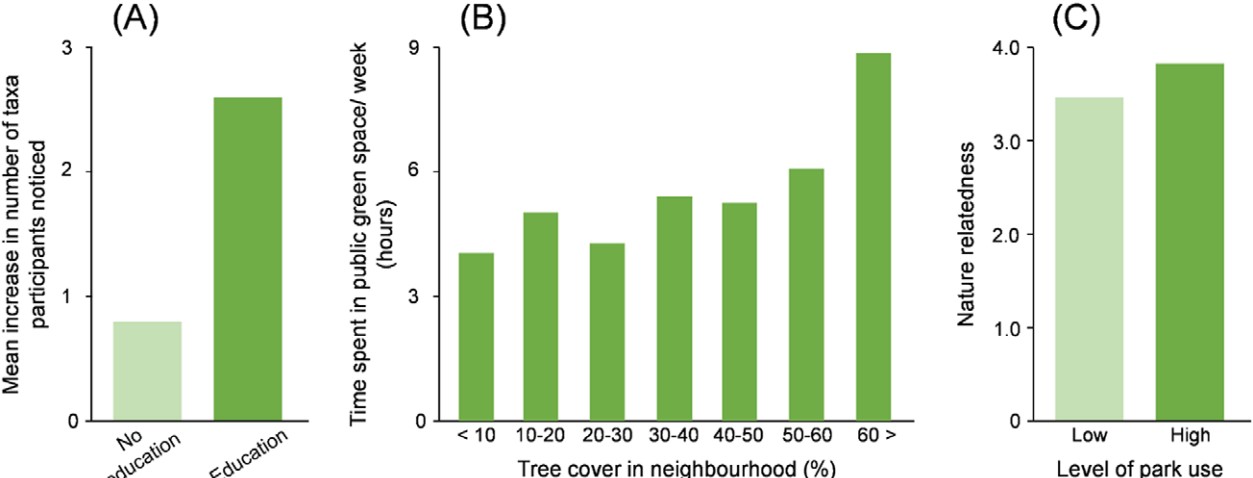

**Figure 3.** Empirical evidence demonstrating the role of (A) capability, (B) opportunity and (C) motivation in determining personalised ecology. Plots show: (A) effects of an educational program aimed at increasing children's species identification ability on the number of animal and plant taxa children noticed on the way to school (Lindemann-Matthies, 2002); (B) association between neighbourhood tree cover and time spent in public greenspace (Shanahan et al., 2017); and (C) differences between park users (based on time spent in parks) in terms of their emotional connection to nature (measured by the Nature Relatedness scale; Lin et al., 2014).

### (i) Capability

Capability is an individual's capacity to engage in interactions with nature. It has two components, physical capability and psychological capability. Physical capability includes the ease with which one can move around and the extent to which one has sufficient sensory abilities to detect particular species, for example, being able to see birds or hear birdsong. Psychological capability includes knowledge, skills, stamina and confidence. The component that has attracted the most attention is skills such as the ability to recognise particular species (Figure 3A; Lindemann-Matthies, 2002).

Arguably, biodiversity conservation has been heavily fixated on improving a rather narrow conception of capability – assuming that education about the nature around you will improve your ability to access it, and willingness to protect it (Thomas-Walters et al., 2023). This is despite various studies finding that education and knowledge, by themselves, are relatively poor predictors of connection to nature (e.g., Barragan-Jason et al., 2022) and pro-nature behaviours (e.g., Knapp et al., 2021).

Capability might be improved by (i) improving ways for less physically able people to interact with nature (e.g., via views from windows, improved access to greenspaces); (ii) equipment that enables people to overcome or reduce sensory limitations in interacting with nature (e.g., vision and acoustic systems); (iii) accessible tools and learning that help improve psychological capability; and (iv) guides (particularly people rather than signage) who can facilitate and explain nature interactions to visitors to sites.

### (ii) Opportunity

Opportunity is all of the factors in a person's environment that make interactions with nature possible. It has two components, physical opportunity and social opportunity. Physical opportunity includes the availability of nature in a person's environment with which they can interact (Figure 3B; Shanahan et al., 2017). Social opportunity includes family values, social norms and public safety. Attention has particularly focused on the role of physical opportunity in personalised ecology, and the extent to which people have adequate or appropriate access to nature in their immediate

neighbourhood or more widely. Indeed, some organisations have established targets for the availability of local greenspace, such as at least 0.5 ha within 200 m, 2 ha within 300 m and 10 ha within 1 km, all within a 15 min walk from home (Natural England, 2022). Social opportunity, on the other hand, has received much less attention in discussions on how to promote people's use of nature, except for some particular cases such as children's use of local nature (Button et al., 2020). However, recent studies suggest that the influence of social opportunity on personalised ecologies is comparable to – and sometimes stronger than – that of physical opportunity (e.g., Soga et al., 2018; Van Truong et al., 2022).

Opportunity might be improved by: (i) improving the availability and accessibility of local nature, including in the vicinity of both home and work places; (ii) improving the ability of nature to move amongst greenspaces, influencing both species' population sizes and the potential for human–nature interactions; (iii) physically 'greening' buildings and their immediate surroundings, through green roofs and walls, gardens, etc.; (iv) improving transport systems to enable better access to nature sites; (v) changing and challenging values, social norms and normative beliefs around nature interactions (e.g., through community engagement, community champions and role models); (vi) improving safety of the local environment (e.g., improving road safety, reducing crime, controlling the abundance of wild animals that can have severe negative health impacts on people); and (vii) providing more dedicated time for nature interactions (e.g., built into work/school schedules).

### (iii) Motivation

This is the set of brain processes that energise and direct behaviour. Its two components are: automatic motivation and reflective motivation. Automatic motivations are unconscious responses, such as emotional reactions, whilst reflective motivations are more cognitive and purposeful, such as intentions (Figure 3C; Lin et al., 2014). In the field of human–nature interactions, the importance of motivation (particularly automatic motivation) in promoting personalised ecologies has long been recognised, and indeed studies show that it is often the most impactful factor in predicting the

quantity and quality of those interactions (e.g., Lin et al., 2014; Soga and Akasaka, 2019). Of course, motivation is likely to be improved by enhanced personalised ecologies (see earlier discussion on biophilia), implying that there exists a bidirectional relationship between motivation and personalised ecology.

Motivation might be improved by: (i) green social prescribing, which can provide an incentive for reflective motivation and intentions to interact with nature; (ii) nature-based educational programs in educational institutions (e.g., schools, museums) that can help to increase connection to nature, and therefore automatic motivation; and (iii) nature-oriented television and internet programs (e.g., nature documentaries), and social media that promote people's desire to experience nature.

Strategies to improve opportunity, motivation and capability do not work independently but are interrelated in many ways. For example, providing nature-based education in schools can help to increase all of the three drivers (capability: ability to notice wildlife; opportunity: ensuring time to interact with nature; motivation: nature connectedness). Improving emotional connection to nature, through recreational and educational programs, is also known to be closely related to enhanced psychological wellbeing (Pirchio et al., 2021), suggesting that it may help to increase psychological capability.

## Conclusions

Many factors shape people's behavioural decisions, small or large, which collectively determine the future of biodiversity. People's personalised ecologies are a central factor that may act directly (impacting nature during people's interactions) or indirectly (influencing mediating factors such as attitudes, nature connectedness and normative beliefs). This raises the potential of a virtuous cycle whereby improving personalised ecologies encourages demand for improved biodiversity, at a time when a high proportion of the global population's interactions with nature are extremely constrained, and becoming poorer.

**Open peer review.** To view the open peer review materials for this article, please visit http://doi.org/10.1017/ext.2023.15.

**Data availability statement.** Data availability is not applicable to this article as no new data were created or analysed in this study.

**Acknowledgements.** We are grateful to three anonymous reviewers for their thoughtful comments.

**Author contribution.** Conception and design of work: K.J.G., B.B.P. and M.S.; Drafting and revising: K.J.G., B.B.P. and M.S.

**Financial support.** K.J.G. and B.B.P. were supported by the Natural Environment Research Council funded 'Renewing biodiversity through a people-in-nature approach (RENEW)' project (NE/W004941/1). M.S. was supported by the Japan Society for the Promotion of Science (Grant Nos. 20H04375 and 23H03583).

**Competing interest.** The authors declare no competing interest exists.

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
