## [Reviewer Report]

I’ve found the paper an interesting read and overall I only have very minor comments for improvement. However, I have to disclose that while I have experience in conservation research, I’m not familiar with the literature of personalised ecology and in general behavioural science applied to conservation, so I’m unable to judge the originality and correctness of the content. Nonetheless, below you will find my suggestions. 

The document did not have line numbers, so I indicated the page and the whole sentence where needed.

page 4: “In general, personalised ecologies are dependent on opportunity (e.g., the local presence and abundance of species), motivation (e.g., emotional affinity with nature) and capability (e.g., ability to see or hear particular species) (Dallimer et al., 2014; Soga and Gaston, 2022”. 

And previous experiences/Instruction? Both motivation and capabilities ultimely depend in large part on how a person has been raised. These interests can be shaped by family, friends, local culture, etc.

page 4: “a phenomenon known as extinction of experience”. I’d not use “known” since this was used in a few papers, perhaps “named”, “labeled” or “referred to”.

page 4: sentence starting with “On the other hand, international...” specify this relates to a minority of humanity (wealthy people that are already much attracted by nature, i.e. a minority of a minority).

“(ix) Biodiversity monitoring” I suggest changing this to “citizen science”, which clearly can be a form of biodiversity monitoring in some contexts (with many “if”s). 

Page 8: “assuming that education about the nature around you will improve your ability to access it.” your ability or your interest? I guess this has to do with the definition of capabilities and my previous comment on education. But see next comment. 

Page 8: “This is despite various studies finding that education and knowledge are poor predictors of connection to nature (e.g., Barragan-Jason et al., 2022) and pro-nature behaviours (e.g., Knapp et al., 2021).” This seems to contradict the statement in page 5 “...in subsequent writings he emphasized that it is a complex set of learned behaviours, i.e. a disposition that is reinforced, amplified and expressed through human culture (Wilson, 1993). This is now supported by empirical evidence (Figure 2A; Collado et al., 2013; Vanderstock et al., 2022).” It requires a comment from the authors. 

Page 9: “Psychological capabilities” the term is not adequate in my opinion, it seem to refer to very different skills (e.g. self confidence or similar). I checked and the cited reference (Lindemann-Matthies) did not use it. I’d suggest something like “wildlife identification skills”, or more general, e.g. “nature-related skills”.

Page 10: “Motivation can be improved...” you may also consider social media and influencers in moderns times.

Fig. 1. Can you highlight the negative and positive consequences (e.g. different colours, position, etc..). I’d also change “Biodiversity monitoring” to “Citizen science”

Fig. 2. The figure requires going back and forth between each panel and the legend to be interpreted. It would be better if you could make it more intuitive at a first sight. e.g. in D i suggest replacing “absent/present” with “with recreational activities” “without recreational activites”. In F you may specify somehow that before/after refers to experience in urban green park. etc. 

You could do this by adding a title. e.g. in A “Participation in a nature-based recreational program”

This is not much different than what you did in Fig. 3C for example.

---

## [Reviewer Report]

This manuscript deals with the relationship people have with nature and how this relationship impacts on the future of global biodiversity. Somewhat sadly, this topic is a great fit to the focus of this journal. The manuscript is well written and topical, building upon a number of recent publications on the same issue by the submitting authors (taking the conceptual framework from Soga & Gaston 2022) and looking more specifically at the consequences for biodiversity.

The primary issue I have with the manuscript in its current form is the overly western-centric framing on a topic that I would like to think has far broader relevance to non-western cultures. Granted, much of this centricity is implicit in the text, but I see value in considering a more explicitly diverse approach. At a number of points throughout the manuscript, this limitation is touched on or wider diversity is recognised. For example “nature connectedness varies dramatically amongst people and societies” (p 5) and “traditional management practices of local ecosystems” (p 7). Interestingly, in Soga & Gaston (2022), this framing was recognised as a major caveat to the underlying analysis: “…human-nature interactions can vary substantially across societies with different cultural backgrounds…”. 

My point is that I think this manuscript would make a far more valuable contribution to the literature if it was to give greater consideration to how other non-western cultures fit within the outlined concept (i.e. that summarised in Fig. 1) and, in turn, how their different perspectives influence consequences for biodiversity. The reason I see this change as a priority is that much of the global biodiversity that is imperilled has a future that is strongly influenced by the personalised ecology of non-western communities. In particular, I see merit in giving greater consideration to the role of culture, particularly Indigenous culture, in refining the framing. For example, on one hand culture is likely to be a significant driver of personalised ecology for some communities in a way that doesn’t fit neatly into the existing boxes of the COM-B system. Equally, culture is inter-woven through many of the consequences of personalised ecology in a way possibly implied by the ‘feedback loops’ mentioned in the caption for Fig. 1.

A secondary issue is one of clarity – what is meant exactly by biodiversity, and how does this relate to non-native species? Again, implicitly, biodiversity appears to be used to refer to native biological diversity. However, some of the examples given, particularly in regard to ‘nature’ and ‘greenspaces’, is non-specific in regard to native status. Given that there is a well-established link between species introductions into urban areas (often for horticulture and as pets) becoming threatening non-native escapees, is this a missed opportunity to more explicitly recognise that there are possible downsides to these aspects of personalised ecology? This link might fit nicely with strategies to mitigate such risks that are already discussed, such as negative aspects of nature engagement, shifting baselines and ecological literacy.

Minor issues (noting no line numbering provided):

P 6, para 3: please change ‘invasive’ to ‘non-native’ or ‘alien’; the term invasive is frequently misunderstood/misused and it is more commonly accepted in the invasion literature to refer to a rapid increase in distribution, rather than non-native status.

P 6, para 3: isn’t the potential disturbance equally relevant to plants as it is to animals? If so, I suggest changing ‘wildlife’ to ‘flora and fauna’ (or something equivalent) in the last sentence.

P 8, para 3: the statement “biodiversity conservation has been heavily fixated… …improve your ability to access it” is a bold claim, and one that I think either needs stronger support from the literature via citation, or a more nuanced explanation as to why it is a sound assertion.

P 10, para 4: is ‘demand for greater biodiversity’ the most appropriate goal to list here? In the strict sense of the term, communities with more numerous species are not necessarily what we should be striving for if biodiversity conservation (i.e. determining the future of biodiversity) is the goal.

Figure 1: I suspect there is a more elegant way of presenting the concept than the structure currently presented. I would encourage the authors to think outside the box to try and include a version that incorporates the mentioned feedback loop and considers my earlier feedback on the matter of culture.

Figure 1: would it be clearer to say ‘drivers’ rather than ‘causes’ in the second sentence?

Soga, M., Gaston, K.J. Towards a unified understanding of human–nature interactions. Nat Sustain 5, 374–383 (2022). https://doi.org/10.1038/s41893-021-00818-z

---

## [Editor Report]

We have now received three reviews for this submission. All found the topic interesting and important. However, two reviewers raised some issues with the current version of the manuscript. In particular, please address comments on the need to give greater consideration to how non-western cultures fit within your proposed concept. And how they can offer different perspectives to affect biodiversity conservation. As such, I would like to invite the authors to revise their manuscript.

---

## [Reviewer Report]

I’m satisfied with the response given and the revised version of the paper, I can now recommend publication.